# Multidimensional Analysis of Quality of Life in Patients with Chronic Non-Cancer Pain and Short- and Long-Term Intrathecal Analgesic Therapy

**DOI:** 10.3390/healthcare12181870

**Published:** 2024-09-18

**Authors:** Manuel Alejandro Sánchez-García, Bernardino Alcázar-Navarrete, Manuel Cortiñas-Saenz, Nicolás Cordero Tous, Rafael Gálvez Mateos

**Affiliations:** 1Pain Unit, Anesthesiology and Resuscitation Department, Hospital Universitario Virgen de las Nieves, Avda. Fuerzas Armadas, sn., 18014 Granada, Spain; mcortinassaenz@gmail.com (M.C.-S.); rafaelgalvez@hotmail.com (R.G.M.); 2Medicine Department, University of Granada, 18011 Granada, Spain; balcazarnavarrete@gmail.com; 3Pulmonology Department, Hospital Universitario Virgen de las Nieves, 18014 Granada, Spain; 4Neurosurgery Department, Hospital Universitario Virgen de las Nieves, 18014 Granada, Spain; voraxanimatur@hotmail.com

**Keywords:** long-term spinal pain, chronic low back pain, intrathecal infusion, intrathecal infusion systems, intrathecal drug delivery systems, intrathecal pumps, chronic non-cancer pain, chronic non-malignant pain, morphine, ziconotide, health-related quality of life

## Abstract

**Background:** Intrathecal drug delivery (IDD) is part of the fourth analgesic step. Evidence on the quality of life of patients with refractory chronic non-cancer pain (CNCP) using these devices and their long-term outcomes is scarce. This study aims to evaluate patients with IDD to assess their HRQoL. Additionally, the study seeks to understand the patients’ satisfaction with the treatment and changes in pain magnitude over time. **Methods:** Adult patients with CNCP and intrathecal drug delivery systems (IDDS) were included. The study population was divided into two groups: less than and more than 15 years of treatment. HRQoL was analyzed using validated questionnaires. Pain reduction was assessed using the visual analog scale (VAS), and treatment satisfaction was evaluated using the Patient Global Impression of Improvement scale. **Results:** The results indicate a poor HRQoL in IDD patients, with better scores in the group with ≥15 years of treatment. Pain reduction was similar in both groups, and patients reported a positive satisfaction level with the treatment. **Conclusions:** HRQoL in CNCP patients is severely affected. Long-term IDD patients have a similar or even better HRQoL in some respects compared to those with shorter follow-ups. IDD patients experienced pain reduction, with most feeling better or much better.

## 1. Introduction

Chronic pain is a severe, frequent, and current socio-health problem that leads to significant deterioration in the quality of life for patients. It is among the leading causes of suffering and disability, with an estimated prevalence of about 20% in Europe and 25–40% in Spain. A major socio-political, medical, and technological effort is necessary to tackle this problem [1,2,3,4].

The World Health Organization (WHO) defines health-related quality of life (HRQoL) as an individual’s perception of their position in life, in the context of their culture and value system and concerning their goals, expectations, standards, and concerns. This broad concept is influenced by the patient’s physical and mental state, social relationships, and environment [5].

Intrathecal drug delivery (IDD) is part of the fourth analgesic step for managing these patients’ pain and has been practiced in humans since 1979 [6]. In 1991, the US Food and Drug Administration (FDA) approved the administration of drugs, including morphine, baclofen, and ziconotide, via this route. This technique involves administering drugs into the subarachnoid space to modulate nociceptive information at this level, providing more selective, effective analgesia with lower doses and fewer side effects [7,8,9,10,11]. It is reserved for patients with refractory chronic cancer pain and chronic non-cancer pain (CNCP) who do not respond to other treatments or experience intolerable adverse effects with systemic opioids. Common indications for CNCP include persistent low back pain syndrome and complex regional pain syndrome [11,12,13,14,15,16,17,18,19].

There is extensive literature on the efficacy of this treatment in cancer pain patients, typically focusing on pain severity [3,9,20,21,22,23] However, there is controversy about its effectiveness in CNCP patients, and few studies have assessed these patients’ quality of life [11,13,23,24,25,26,27,28], especially from a multidimensional and long-term perspective [11,15,29].

This study aims to evaluate a group of patients with severe and refractory CNCP undergoing IDD at our center to assess their quality of life using validated scales from a multidimensional perspective, comparing results between those with shorter and longer treatment durations. Additionally, the study seeks to understand their satisfaction with the treatment and changes in pain magnitude over time.

## 2. Methodology

### 2.1. Design

A single-center cross-sectional observational study was conducted.

### 2.2. Study Population

Consecutive adult patients with CNCP under the Pain Unit of Hospital Universitario Virgen de las Nieves (Granada, Spain), treated with intrathecal drug delivery systems (IDDS) (SynchroMed II^®^, Medtronic, Minneapolis, MN, USA or Tricumed^®^, Tricumed Medizintechnik GmbH, Kiel, Germany) for morphine and/or ziconotide infusion, were included from September 2023 to March 2024. Inclusion criteria were refractory CNCP, treatment with IDDS for more than 3 months, being of legal age, and agreeing to participate in the study by signing informed consent. Exclusion criteria included inability to respond to study questionnaires, less than 3 months of treatment, chronic cancer pain, and refusal to participate. Two main groups were differentiated: 15 years or more with IDD and less than 15 years with IDD.

### 2.3. Study Objectives

The primary objective was to conduct a multidimensional analysis of the HRQoL of CNCP patients using IDDS, dividing the study population into those with more than and those with less than 15 years of treatment. Secondary objectives included assessing their satisfaction with the treatment and the percentage of pain reduction with the implanted device over time.

### 2.4. Variables

Baseline demographic and anthropometric data (age, sex, height, BMI, and education level) were collected from all participants. Anthropometric measurements were performed with a stadiometer and a scale. Additional clinical data included CNCP etiology, type of pain, years with IDDS, previous interventional techniques, differences between analgesic consumption before and after the implant, pre-implant dysthymia, and the type of drug used in the infusion device.

### 2.5. Questionnaires

We assessed the differences in pain levels before and after the device implant using the visual analog scale (VAS) by reviewing patients’ medical records. For the health-related quality of life (HRQoL) assessment, we administered several validated self-reported health questionnaires with their validated Spanish version on the review day.

SF-12 [30]: A self-reported, 12-item questionnaire that analyzes HRQoL in two domains: mental (MCS-12) and physical (PCS-12), with a mean value of 50 for each domain.

Brief Pain Inventory—Short Form (BPI-SF): The short form of this self-reported inventory consists of 11 questions that assess pain interference with quality-of-life aspects, scored from 0 to 10, resulting in an overall score from 0 to 110 [31,32].

European Quality of Life—5 Dimensions (EQ-5D-5L) [33,34,35,36]: This self-reported instrument evaluates five relevant life aspects with five severity levels and allows for comparison with the reference population index, which ranges from 0 to 1.

Additionally, we used the Patient Global Impression of Improvement (PGI-I) scale to assess the patient’s overall impression regarding the device’s application, results, and tolerability, scored from 1 to 7 (1 being “much improved” and 7 being “much worse”). On the consultation day, patients also self-reported their health status using the EQ-5D VAS, ranging from 0 to 100 [37].

### 2.6. Statistical Analysis

Data are presented using measures of central tendency and dispersion (mean and SD) for continuous variables and as a number (and percentage of total) for ordinal variables. Normal distribution was evaluated using the Kolmogorov–Smirnov test and histograms. Comparison of normally distributed continuous variables was performed using Student’s *t*-test or ANOVA. Non-normally distributed continuous variables were compared using non-parametric tests like Mann–Whitney U or Kruskal–Wallis. Ordinal variables were compared using the X^2^ test. All statistical analyses were performed using Jamovi version 1.6 (The Jamovi Project 2021). A *p*-value < 0.05 was considered statistically significant.

### 2.7. Ethical Aspects

The study was conducted in accordance with the Helsinki Declaration (Fotaleza revision, October 2013) and the Organic Law 3/2018 of 5 December on Personal Data Protection and Guarantee of Digital Rights and was authorized by the ethics committee of the center (1666-N-23, 1 January 2023). All participants gave their consent to participate in the study by signing the informed consent.

## 3. Results

A total of 88 patients were screened; of these, 84 were finally included (Figure 1), 22 (26.2%) with 15 years or more of IDD for CNCP and 62 (73.8%) with less than 15 years. Ages ranged from 34 to 83 years (mean age 57 years). The general description of our study population and the differences between the major groups (more than or less than 15 years with IDD) are shown in Table 1. This sample maintained similar characteristics over time, with no significant differences between groups except for age at implantation and the use of interventional techniques, which were significantly lower in patients with 15 years or more with IDD.

The main indication for implantation was persistent low back pain syndrome, followed by generalized pain. The most predominant pain type was mixed (with nociceptive and neuropathic characteristics), although percentages were not far from the rest.

There was a decrease in opioid consumption, with more than half not consuming opioids at the time of data collection, and only 20% continuing with potent opioids.

Focusing on the HRQoL data analysis (Figure 2), it is observed that the obtained scores are below the average numeric values, although if separated by groups, HRQoL was numerically better in the 15-years-or-more treatment group, with significant differences in BPI-SF, EQ-5D VAS, and the physical score of SF-12 (PCS-12).

Figure 3 shows the global improvement of about 40% in VAS among patients undergoing IDDS. Overall, patients started with an average VAS of 9 and reduced their pain by 3–4 points with treatment. No differences in improvement were found between groups.

In the analysis of patient satisfaction with the treatment using the PGI-I survey, most patients reported improvement with treatment (with only two patients worse at the time of the study), with numerically higher percentages of patients considering themselves much or much better (68.2% of those with 15 years or more of IDD and 45.2% of those with less than 15 years) (Table 2).

No differences were found in HRQoL improvement related to morphine or ziconotide administration or device type between groups.

## 4. Discussion

Refractory CNCP causes significant limitations and suffering for those affected, impacting all aspects of their lives physically and mentally. IDDS offers improvements in patient-reported outcomes measures (PROMs) that seem to be sustained long-term.

The study shows that the HRQoL in patients, regardless of the questionnaire used, was not good. Despite this, results indicate that IDD patients with 15 or more years of treatment have a better HRQoL measured by BPI-SF, PCS-12, and EQ-5D VAS than those with less than 15 years of treatment, with no significant differences in other HRQoL questionnaires used in the study.

Regarding pain control results with treatment, patients showed VAS reductions of 3–4 points in pain, nearly halving their pain from the onset of IDD, leading to significant HRQoL improvements with no differences between groups.

Similarly, these patients achieved a positive treatment satisfaction level, with the majority feeling better after the treatment, supporting the use of these devices for refractory CNCP treatment.

Our study highlights the devastating effect of CNCP on HRQoL. Our data suggest that CNCP patients have mean SF-12 scores below those of the reference population [38] in both physical (PCS-12) and mental (MCS-12) components, aligning with Thimineur et al.’s study [26]. Analyzing the EQ-5D-5L results, we observe that these patients’ physical and mental HRQoL is deteriorated compared to their reference population (0.89 in Spain and 0.88 in Andalusia) [35,38]. Regarding EQ-5D VAS scores, although below average, significant differences were found between the two study groups, being better in those with 15 or more years of treatment.

In terms of the BPI-SF, despite the overall poor scores, we see higher values in those with less than 15 years of treatment, with a slight improvement in those with 15 or more years of treatment, indicating a high degree of pain and daily activity limitation that slightly improves over time. Although the level could be discussed, we chose the cut-off of over 15 years due to the limited literature available on this subject, understanding that from this timeframe onwards, patients are considered to be undergoing very long-term treatment.

Pain improvement, according to VAS scores, although slightly higher than that of the reviewed studies, aligns with the systematic review results by Falco et al. [25], Hayek et al. [3], and Patel et al. [23], showing limited evidence of the treatment’s effectiveness.

Patient satisfaction with treatment, as measured by PGI-I, was positive, supporting Deer et al.’s findings [28]. More patients in the more-than-15-years group reported feeling “much better,” supporting the long-term use of this technique, possibly due to patient adaptation and treatment optimization over time.

Persistent low back pain syndrome is the most common etiological diagnosis, being the primary indication for IDD, consistent with analyzed studies [11,12,13,16].

Noteworthy is the reduction in opioid consumption among these patients, leading to decreased physical and psychological adverse effects associated with opioid use, consequently improving the HRQoL.

Our study has strengths such as a multidimensional approach to assessing patient HRQoL, combining most validated HRQoL scales with pain and treatment satisfaction assessments, and providing a global view of these patients’ situations. Additionally, a significant number of our patients had 15 or more years of IDDS, a scarcely evidenced aspect in the literature regarding long-term efficacy and safety.

Our study also has limitations, such as not being a prospective study that would have allowed us to observe HRQoL improvement evolution relative to previous levels, aligning with Narváez et al. [24], Deer et al. [28], and Hamza et al. [27]. Other weaknesses include being a single-center study, limiting extrapolation to the global population with IDDS, and a limited sample size, possibly preventing us from detecting significant differences in study variables. However, the consecutive sampling reduces the risk of selection bias.

## 5. Conclusions

Refractory CNCP significantly limits and causes suffering to those affected, impacting all aspects of their lives physically and mentally. Overall, patients with IDDS experienced pain reduction with treatment, feeling mostly better or much better after device implantation. These findings suggest a HRQoL improvement with IDD from both physical and mental perspectives, despite scores still being lower than those of the reference population, associated with pain reduction and positive treatment satisfaction. These data indicate that, despite the lack of firm evidence supporting IDD in CNCP patients, they would benefit from this type of treatment, showing both short- and long-term benefits where even HRQoL parameters seem to improve. Periodic HRQoL assessment of these patients is necessary to obtain a multidimensional view of their process and to evaluate its evolution.

## Figures and Tables

**Figure 1 healthcare-12-01870-f001:**
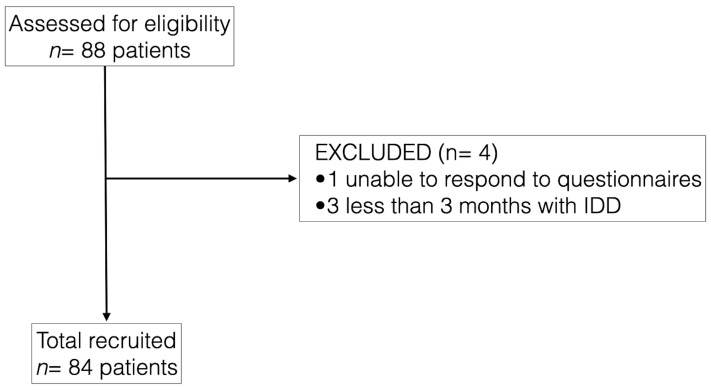
Flow chart of patient selection for the sample.

**Figure 2 healthcare-12-01870-f002:**
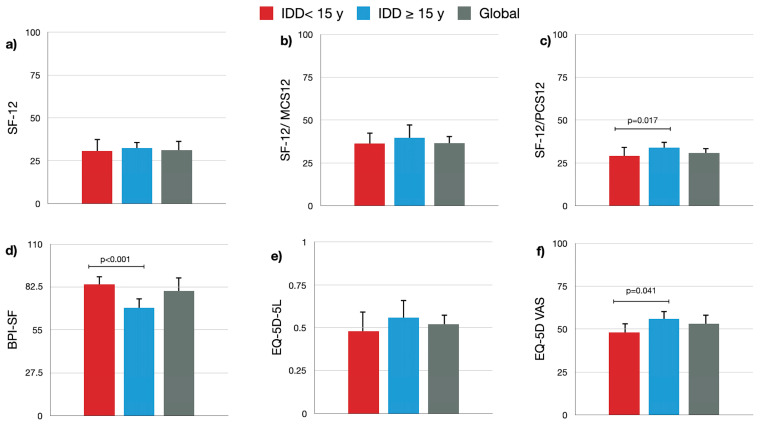
Health-related quality of life (HRQoL) questionnaires among patients receiving IDD for CNCP for ≥15 years (blue bars), <15 years (red bars), and global (grey bars). Mean (SD) results for (**a**) SF-12 questionnaire; (**b**) MCS-12 questionnaire; (**c**) PCS-12 questionnaire; (**d**) BPI-SF questionnaire; (**e**) EQ-5D-5L index questionnaire; (**f**) EQ-5D VAS questionnaire.

**Figure 3 healthcare-12-01870-f003:**
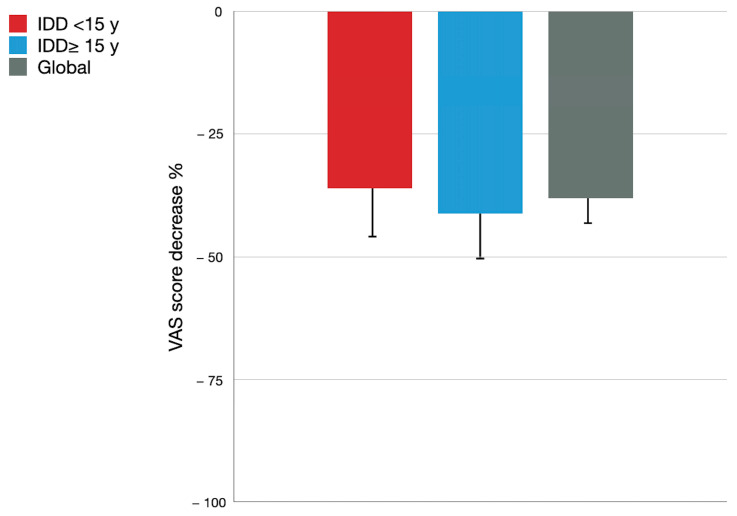
VAS score relative reduction among patients with IDD for CNCP for ≥15 years (blue bars), <15 years (red bars), and global (grey bars) expressed as mean (SD).

**Table 1 healthcare-12-01870-t001:** General overview of study population and differences between those patients included with more than or less than 15 years of IDD. All the data are expressed in mean (±SD) or *n* (% of entire column).

	Global (*n* = 84)	<15 y with IDD (*n* = 62)	≥15 y with IDD (*n* = 22)	*p*-Value
**Age**	56.8 ± 9.8	56.4 ± 10.8	58.2 ± 6.3	0.446 ^1^
**Sex, male**	53 (63.1%)	40 (64.5%)	13 (59.1%)	0.651 ^2^
**Height, cm**	168.9 ± 8.8	168.8 ± 9.4	169.4 ± 7.0	0.790 ^1^
**BMI previous to IDD**	28.5 ± 5.1	28.3 ± 5.1	29.0 ± 5.2	0.601 ^1^
**Years with IDD**	8.9 ± 6.3	5.8 ± 4.0	17.5 ± 2.2	<0.001 ^1^
**Age at IDD**	48.0 ± 11.1	50.6 ± 11.4	40.7 ± 5.9	<0.001 ^3^
**Pre-IDD obesity (BMI > 30)**	30 (35.7%)	21 (33.9%)	9 (40.9%)	0.554 ^1^
**Educational Level**				0.003 ^2^
**Compulsory Schooling or Lower**	54 (64.3%)	40 (64.5%)	14 (64.2%)	
**Vocational Training**	18 (21.4%)	13 (21.0%)	5 (22.7%)	
**High School or University Education**	13 (14.3%)	9 (14.5%)	3 (13.6%)	
**Diagnosis**				0.626 ^2^
**Persistent Low Back Pain Syndrome**	60 (71.4%)	44 (71.0%)	16 (72.7%)	
**Complex Regional Pain Syndrome**	6 (7.1%)	5 (8.1%)	1 (4.5%)	
**Generalized Pain**	15 (17.9%)	10 (16.1%)	5 (22.7%)	
**Non-Surgical Low Back Pain**	3 (3.6%)	3 (4.8%)	0 (0.0%)	
**Type of Pain**				0.955 ^2^
**Somatic Nociceptive**	29 (34.5%)	21 (33.9%)	8 (36.4)	
**Neuropathic**	21 (25.0%)	16 (25.8%)	5 (22.7)	
**Mixed (Nociceptive/Neuropathic)**	34 (40.5%)	25 (40.3%)	9 (40.9)	
**Performance of Interventional Techniques Prior to Implantation**	56 (66.7%)	47 (75.8%)	9 (40.9%)	0.003 ^2^
**Analgesic Consumption Prior to Implantation**				0.549 ^2^
**Non-Opioid Analgesia**	1 (1.2%)	1 (1.6%)	0 (0.0%)	
**Strong Opioids**	83 (98.8%)	61 (98.4%)	22 (100.0%)	
**Current Opioid Consumption**				0.904 ^2^
**None**	46 (54.8%)	34 (54.8%)	12 (54.5%)	
**Weak Opioids**	20 (23.8%)	14 (22.6%)	6 (27.3%)	
**Strong Opioids**	18 (20.2%)	14 (22.6%)	4 (18.2%)	
**Pre-Implant Dysthymia**	58 (69.0%)	45 (72.6%)	13 (59.1%)	0.240 ^2^
**Intrathecal Drug**				0.454 ^2^
**Morphine**	76 (91.7%)	56 (90.3%)	21 (95.5%)	
**Ziconotide**	8 (9.5%)	6 (9.7%)	1 (4.5%)	
**VAS**				
**Initial**	9.4 ± 0.8	9.4 ± 0.9	9.4 ± 0.7	0.960 ^3^
**Current**	5.9 ± 1.8	6.0 ± 1.8	6.0 ± 1.7	0.312 ^3^

IDD: intrathecal drug delivery. SD: standard deviation. BMI: body mass index. VAS: visual analog scale. All the data are expressed in mean (±SD) or *n* (% of entire column). ^1^ Linear model ANOVA. ^2^ Pearson’s Chi-squared test. ^3^ Mann–Whitney U test.

**Table 2 healthcare-12-01870-t002:** Patient Global Impression of Improvement (PGI-I) Scale.

	<15 y with IDD (*n* = 62)	≥15 y with IDD (*n* = 22)	Total
**Very much improved**	5 (8.1%)	4 (18.2%)	9 (10.7%)
**Much improved**	23 (37.1%)	11 (50.0%)	34 (40.5%)
**Minimally improved**	32 (51.6%)	7 (31.8%)	39 (46.4%)
**Minimally worse**	1 (1.6%)	0 (0.0%)	1 (1.2%)
**Much worse**	1 (1.6%)	0 (0.0%)	1 (1.2%)
**Total**	62	22	84

*p* = 0.367.

## Data Availability

Data are contained within the article.

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
