# Peer review of "Multidimensional Analysis of Quality of Life in Patients with Chronic Non-Cancer Pain and Short- and Long-Term Intrathecal Analgesic Therapy"

_healthcare, 2024, doi:10.3390/healthcare12181870_

Round 1
Reviewer 1 Report
Comments and Suggestions for Authors
1. Why only one center is selected for the study? the sampling will be purposive sampling in this case
2. Mention the period of study
3. Mention ethical permission letter reference number with date
4. brief how the demographic (height and weight) data was collected. what instruments used.
5. Methodology needs improvement. explain in brief your procedures and questionnaires so that others can repeat the study with ease
6.Replace very old ( more than 10 years) references with recent one
Comments on the Quality of English Language
please check minor grammatical errors in the methodology section
Author Response
RESPONSE TO REVIEWER #1
- Why only one center is selected for the study? the sampling will be purposive sampling in this case
We thank R#1 for this comment. This is a single-center study that comprises all the patients at our unit that are under IDD. This is recognized in the discussion section (page 9, lines 205- 207). However, we have reinforced this information in the methods section and in the discussion section.
“Consecutive adult patients…” (page 2, line 65)
“However, the consecutive sampling reduces the risk of selection bias” (page 9, line 208).
- Mention the period of study
Patients were recruited from Sept’23 to March’24 as it is stated in the document (Page 2, line 69)
- Mention ethical permission letter reference number with date
Sorry for not having included this in the manuscript.
We have included the date in the ethical aspects: 1666-N-23, Jan 1st 2023 (Page 3, line 115)
- brief how the demographic (height and weight) data was collected. what instruments used.
We would like to thank R#1 this comment. We have modified thee manuscript that now reads as
“Anthropometric measurements were performed with a stadiometer and a scale” (Page 2, line 82)
- Methodology needs improvement. explain in brief your procedures and questionnaires so that others can repeat the study with ease
Thank you for your comment. We have clarified our methodology to ensure that others can easily replicate the study. We have modified the methods section that now reads as:
We assessed the differences in pain levels before and after the device implant using the Visual Analog Scale (VAS) by reviewing patients' medical records. For the Health-Related Quality of Life (HRQoL) assessment, we administered several validated self-reported health questionnaires with their validated spanish version on the review day:
SF-12 (39): A self-reported 12-item questionnaire that analyzes HRQoL in two domains: mental (MCS-12) and physical (PCS-12), with a mean value of 50 for each domain.
Brief Pain Inventory – Short Form (BPI-SF): The short form of this self-reported inventory consists of 11 questions that assess pain interference with quality of life aspects, scored from 0 to 10, resulting in an overall score from 0 to 110 (40-41).
European Quality of Life-5 Dimensions (EQ-5D-5L) (42-46): This self-reported instrument evaluates five relevant life aspects with five severity levels and allows for comparison with the reference population index, which ranges from 0 to 1.
Additionally, we used the Patient Global Impression of Improvement (PGI-I) scale to assess the patient's overall impression regarding the device's application, results, and tolerability, scored from 1 to 7 (1 being "much improved" and 7 being "much worse"). On the consultation day, patients also self-reported their health status using the EQ-5D VAS, ranging from 0 to 100 (47)..
(pages 2&3, lines 88-104)
We hope this clarifies our procedures and ensures that the study can be replicated with ease.
6.Replace very old ( more than 10 years) references with recent one
We would like to thank R#1 for pointing out this issue, We have modified the reference section, excluded the oldest references and included updated references.
Reviewer 2 Report
Comments and Suggestions for Authors
This manuscript describes a cross-sectional analysis of pain and HRQOL reports from a large treatment center in Spain in adults patients after previous placement of intrathecal drug delivery systems (IDDS) for refractory CNCP. The study strengths include the use of well-validated measures and the inclusion of patients with clinical follow-up of over 15 years. The results suggest improvement in pain intensity and general satisfaction with the treatment outcomes, but continued poor HRQOL as would be expected in a cohort with severe chronic pain.
1. Please provide a CONSORT diagram providing information on recruitment as it is unclear how representative the enrolled participants are to their entire patient population.
2. Please provide a clinical or scientific rationale for the main sub-group analysis of < 15 years vs > 15 years from IDDS placement, as this seems quite arbitrary and has little obvious clinical significance.
Author Response
RESPONSE TO REVIEWER #2
This manuscript describes a cross-sectional analysis of pain and HRQOL reports from a large treatment center in Spain in adults patients after previous placement of intrathecal drug delivery systems (IDDS) for refractory CNCP. The study strengths include the use of well-validated measures and the inclusion of patients with clinical follow-up of over 15 years. The results suggest improvement in pain intensity and general satisfaction with the treatment outcomes, but continued poor HRQOL as would be expected in a cohort with severe chronic pain.
- Please provide a CONSORT diagram providing information on recruitment as it is unclear how representative the enrolled participants are to their entire patient population.
We would like to thank R#2 for this point. We have incorporated a CONSORT diagram as figure #1
- Please provide a clinical or scientific rationale for the main sub-group analysis of < 15 years vs > 15 years from IDDS placement, as this seems quite arbitrary and has little obvious clinical significance.
We would like to thank R#2 for this smart comment. We have included a sentence in the discussion section about this sub group analysis that now reads:
“Although the level could be discussed, we chose the cut- off of over 15 years due to the limited literature available on this subject, understanding that from this timeframe onwards, patients are considered to be undergoing very long-term treatment” (pages 8&9, lines 191-194)
Reviewer 3 Report
Comments and Suggestions for Authors
Dear Authors,
My recommendations are primarily minor and aim to improve the clarity and quality of your data presentation.
I suggest dividing subsection 2.4, titled "Variables," into two parts. The first part should present the questionnaires used. Please specify the language of the questionnaire and provide a reference to the paper that validated this questionnaire in the language employed, or indicate the Cronbach’s alpha if you conducted the validation yourself. Under the "Variables" subheading, provide more detailed information about the variables collected, including the sources for demographic and anthropometric data.
In the "Statistical Analysis" subsection, you mention that you used Kolmogorov-Smirnov tests to assess data distribution normality. This test alone is insufficient for justifying parametric analysis. Please also use graphical methods (e.g., Q-Q plots, histograms) to assess normality and conduct homoscedasticity testing. Based on the presentation of your data in Tables 1 and 2, it appears you assumed a normal distribution, which is unlikely given the small sample size and the unhealthy population under study.
In Tables 1 and 2, please specify the measures of central tendency and dispersion used. Also, apply inferential analysis for between-group comparisons in Table 1. Include the p-values for each statistical test in a separate column within the tables and specify which test was used in each case in the table footnotes.
In addition, please add values to Figures 1 and 2.
Author Response
RESPONSE TO REVIEWER #3
Dear Authors,
My recommendations are primarily minor and aim to improve the clarity and quality of your data presentation.
I suggest dividing subsection 2.4, titled "Variables," into two parts. The first part should present the questionnaires used. Please specify the language of the questionnaire and provide a reference to the paper that validated this questionnaire in the language employed, or indicate the Cronbach’s alpha if you conducted the validation yourself. Under the "Variables" subheading, provide more detailed information about the variables collected, including the sources for demographic and anthropometric data.
We have modified the methods section and have included a Variables section (2.4) and a Questionnaires section (2.5).
The validated Spanish version of each questionnaire were delivered to patients. We have included a sentence regarding the comment of R#2.
In the "Statistical Analysis" subsection, you mention that you used Kolmogorov-Smirnov tests to assess data distribution normality. This test alone is insufficient for justifying parametric analysis. Please also use graphical methods (e.g., Q-Q plots, histograms) to assess normality and conduct homoscedasticity testing. Based on the presentation of your data in Tables 1 and 2, it appears you assumed a normal distribution, which is unlikely given the small sample size and the unhealthy population under study.
We will like to thank R#3 for this comment that we had mistaken. We have included the test performed in each analysis and where normality could not be assumed a Mann Whitney U test was performed.
In Tables 1 and 2, please specify the measures of central tendency and dispersion used.
Data is expressed as mean ± SD as stated in Table 1 heading “All the data is expressed in mean (± SD) or n (% of entire column)”
Also, apply inferential analysis for between-group comparisons in Table 1. Include the p-values for each statistical test in a separate column within the tables and specify which test was used in each case in the table footnotes.
We thank R#3 for his statistical recommendations which have improved the overall quality of the manuscript. We have included a p- value column in Table 1 as well as the test in the table footnotes.
All the data is expressed in mean (± SD) or n (% of entire column). 1Linear model ANOVA. 2 Pearson´s Chi- squared test.
In addition, please add values to Figures 1 and 2.
We have added p- values to Fig 1 & 2 (now they are Fig 2 and Fig 3) due to the insertion of a CONSORT diagram suggested by R#2